# Brazilian Food Waste as a Substrate for Bioethanol Production

**DOI:** 10.3390/foods13244032

**Published:** 2024-12-13

**Authors:** Íthalo Barbosa Silva de Abreu, Rayssa Karla Silva, Joyce Gueiros Wanderley Siqueira, Paula Katharina Nogueira da Silva, Jorge Luiz Silveira Sonego, Rafael Barros de Souza, Antonio Celso Dantas Antonino, Rômulo Simões Cezar Menezes, Emmanuel Damilano Dutra

**Affiliations:** 1Research Group on Biomass Energy, Department of Nuclear Energy, Federal University of Pernambuco, Recife 50740-545, PE, Brazil; ithalo.abreu@ufpe.br (Í.B.S.d.A.); joycegwsiqueira@gmail.com (J.G.W.S.); romulo.menezes@ufpe.br (R.S.C.M.); 2Institute of Biological Sciences, University of Pernambuco, Recife 50100-130, PE, Brazil; raayssa.karla@gmail.com (R.K.S.); paulaknogueira@gmail.com (P.K.N.d.S.); rafael.souza@upe.br (R.B.d.S.); 3Biotechnology Graduate Program, Department of Antibiotics, Federal University of Pernambuco, CEP, Recife 50670-901, PE, Brazil; jorge.sonego@ufpe.br; 4Research Group on Imaging, Characterization and Simulation in Porous Media (ICSPM), Department of Nuclear Energy, Federal University of Pernambuco, Recife 50740-545, PE, Brazil; antonio.antonino@ufpe.br

**Keywords:** municipal solid waste, enzymes, acid digestion, biofuels, food waste

## Abstract

Food waste (FW) is a common source of contamination, contaminating both soils and water bodies by releasing greenhouse gases. FW holds great potential for biofuel and bioproduct production, which can mitigate its environmental impact and become a valuable addition to the circular bioeconomy. Therefore, this work aimed to investigate the use of food waste as a substrate to produce fermentable sugars and bioethanol. FW was pretreated by lipid removal. Raw and treated FW was hydrolyzed by amylases. Also, FW was hydrolyzed using sulfuric acid under different residence times (20, 40, and 60 min), sulfuric acid concentrations (0.5, 1.0, and 1.5% v·v^−1^), solid loads (5, 10, and 15% m·v^−1^), and temperatures (111, 120, and 127 °C). The best reducing sugar concentration was obtained at a 1.5% concentration of sulfuric acid and a 15% solid load applied for 1 h at 127 °C. The acid hydrolysis process was more efficient (76.26% efficiency) than the enzymatic one (72.7%). Bioethanol production was carried out as static submerged fermentation, with *Saccharomyces cerevisiae* at 10% (humidity m·v^−1^) being used as the producer and the acid and enzymatic hydrolysates being used as carbon sources. Lipid removal from FW did not influence the acid or enzymatic hydrolytic processes. For fermentation, the highest bioethanol yield was obtained from the acid hydrolysate of raw FW (0.49 kg·kg glicose^−1^). Thus, the processes used were efficient for bioethanol production, presenting alternatives for sustainable food waste destinations and low-cost biofuel production.

## 1. Introduction

Population growth and intense urbanization increase municipalities’ solid residue generation. Also known as Municipal Solid Waste (MSW), it appears to be an issue for environmental management in municipalities. According to the UN, the world generates 2.1 billion tons of solid waste per year [1], with 44% of it being organic waste [2]. An insurmountable amount of MSW is still disposed of irregularly in open areas, which causes environmental, social, and health issues such as bad odor, soil and groundwater table contamination, and the emission of greenhouse gases (GHGs) [3]. MSW’s common legal destinations are landfills or incinerators. Landfills are distant from urban centers, have a short lifespan, require extensive land area, and cause negative environmental impacts, while incineration releases toxic gases, dioxin, and ashes that cause air pollution, a public health issue [4,5]. This waste is vastly heterogeneous and has in its composition recyclable materials and organic matter. Depending on the region that generates it, food waste (FW) comprises between 25.9 and 66.1% of the waste volume [6].

FW is composed mostly of carbohydrates, lipids, proteins, and ashes, which makes it a suitable matter for microbial growth and the consequent release of CO_2_ when under aerobiosis or CH_4_ under anaerobiosis [4]. Thus, these macronutrients are used as substrates to produce bioproducts using microorganisms, such as enzymes [7], fine chemicals [8], and biofuels like biodiesel and bio-oil [9], biogas [10], bioethanol [11], and biohydrogen [12]. This approach to food waste management is a great addition to the circular bioeconomy and can decrease the production cost of these bioproducts [7]. The biotechnological valorization of food waste to produce biofuels and high-value-added products can contribute to the advancement of the circular bioeconomy as it represents the reuse of biological resources, increasing the efficiency of the system [13]. Several advantages can be highlighted in the use of FW from the perspective of the circular bioeconomy, namely, reducing waste, saving natural resources, improving energy efficiency [14], recovering nutrients, and reducing greenhouse gas emissions [15].

The use of biofuels is an increasing trend in bioprocessing agricultural and food waste as they play a major role in replacing fossil fuels, the primary cause of global warming, and achieving clean energy is one of the ONU’s 17 Sustainable Development Goals (SDGs). In addition, the use of FW can help to achieve sustainable cities and communities, another SDG. Bioethanol is the most produced biofuel on a global scale, with about 112 billion L being produced in 2023 alone [16]. However, its production relies on first-generation feedstocks, such as corn and sugarcane, which are food sources [17]. To partially replace traditional food sources, enhance bioethanol production, contribute to the energy transition, and reduce greenhouse gas emissions in the transportation sector, nonconventional substrates—such as lignocellulosic biomass and food waste—are being highlighted as feedstocks for biofuel production [18].

Several chemical, biochemical, physical, and thermal processes are used to extract the best nutrients from FW to serve as substrates to produce bioproducts. This is a trend in sustainable biofuel research, and different methods and arrangements for processing have been widely discussed [19]. Bioethanol production from FW is performed according to the following steps: (1) segregated FW gathering, (2) pretreatment and carbohydrate hydrolysis, (3) the fermentation of monosaccharides, and (4) distillation for ethanol recovery [20]. In these steps, several bottlenecks hold back the bioconversion of FW to biofuels and bioproducts. In the first step, FW must be properly collected and classified, which can be costly. The required pretreatments are used to decrease the particle size and remove chemical and biological contaminants that may intervene in the fermentation process [21]. Carbohydrate hydrolysis needs to be optimized and evaluated to define the best technological route because FW is a particularly heterogeneous matter [22]. The use of the acid hydrolysis route for food waste has certain advantages when compared to enzymatic hydrolysis; it tends to be cheaper since the acids used (sulfuric and hydrochloric) are produced on a large scale and are widely available in different regions. In addition, it has a faster carbohydrate conversion rate. Enzymes have high specificity; therefore, for a heterogeneous substrate such as FW, acid hydrolysis may be the preferable route. Finally, fermentation must be carried out efficiently and lead to broths with more than 4% v·v^−1^ ethanol for effective distillation [23].

To mitigate these bottlenecks, several studies have been conducted on the hydrolysis and fermentation step. For hydrolysis, the use of enzymatic cocktails and diluted acid [24,25] reached efficiencies above 40% in monosaccharide conversion. The use of well-adapted and engineered yeasts is the main approach used to improve the fermentation step in bioethanol production [26,27]. The ability to metabolize sugars into complex substrates varies strongly between different yeast strains. Different types of yeast can have different levels of tolerance to factors such as pH, temperature, and salt concentrations [28,29]. Comparing the fermentative efficiency of different strains makes it possible to identify those that are more efficient in the complete fermentation of food waste hydrolysates, which can increase productivity and reduce costs [30,31]. Recent studies have been carried out to compare and identify the best yeasts for different substrates [32,33].

In this way, this work aimed to use food waste from canteens to produce bioethanol by comparing two different hydrolysis steps: using diluted sulfuric acid and enzymes from commercial enzymatic cocktails and using four industrial and commercial yeast strains.

## 2. Materials and Methods

### 2.1. Food Waste

The FW was collected in September of 2020 from an industrial effluent treatment company serving shopping centers, food distributors, and cafeterias located in the metropolitan area of Recife. The waste was separated by the company and labeled as canteen waste. The visual gravimetric composition of the FW included beans, rice, pasta, vegetables, fruit peels, and meat. After the inspection of the FW, it was milled in a food mill (Triturados ½ HP, Franke, Sapiranga, Brazil), dried in an in an oven with air circulation and renewal (TE-394/1, Tecnal, São Paulo, Brazil) at 65 °C for a week, and milled again, until a granulometry of 20 mesh was obtained, in a mill-type Willey (TE-394/1, Tecnal, São Paulo, Brazil). Samples with a size of 5 g were withdrawn for chemical characterization.

### 2.2. Acidic Hydrolysis

The acid hydrolysis of the FW was carried out under different conditions in which sulfuric acid concentration (0.5%, 1.0% and 1.5% v·v^−1^), reaction time (20, 40 and 60 min), and solids loading (5, 10, and 15%) were combined in an experimental design. The experiments were performed in an autoclave at 127 °C, Erlenmeyers of 500 mL, and a workable volume of 300 mL. The reducing sugars (RS) were measured and were considered as the dependent variable. The best assay was carried out again in triplicate to ensure experimental coherence of the results.

### 2.3. Enzymatic Hydrolysis

The enzymatic hydrolysis of the FW was performed with two enzymatic cocktails, α-amylase (500 U/mL) and amyloglucosidase (260 U/mL), both purchased from Sigma-Aldrich. The experiments were carried out in Erlenmeyer flasks of 250 mL in shaker tables (TE-4200 INCUBADORA, Tecnal, Brazil) for 48 h at 150 rpm and 55 °C. The reaction media were buffered with a phosphate buffer to pH 6.0. The assays had 10% (m·v^−1^) FW and 50 U/g of each enzyme. Sample aliquots were collected at the beginning of the experiment, after 24 h, and after 48 h. The samples were centrifuged and the supernatant used for sugar analysis. Two control experiments were conducted, one with FW and buffer and another with enzyme and buffer.

### 2.4. Ethanol Fermentation

The liquid fraction of the acid hydrolysate was used for media formulation to produce bioethanol. Firstly, the pH of the hydrolysates was adjusted to pH 4.8. The microbial producers were commercial lyophilized yeasts: the *Saccharomyces cerevisiae* JP1 from ethanol [34], two strains used for wine fermentation, TR-313 [35] and E491 [36], and *Saccharomyces cerevisiae* Fleischmann, purchased in a supermarket. The yeasts chosen in this study have widespread commercial and industrial use, so it is of interest to identify their fermentability in a heterogeneous medium such as food waste hydrolysate. The yeasts were activated in YPD medium (10% m·v^−1^ of cells) under constant agitation at 250 rpm for 24 h. For the experiments, 10% (m·v^−1^) of yeast inoculum was added to the hydrolysates and kept at 30 °C for 24 h under static conditions. Sample aliquots were withdrawn at the beginning and at the end of fermentation to quantify reducing sugars, ethanol, and glycerol.

#### Yield Coefficients, Efficiency, and Productivity Calculations

According to the methods of Silva et al., (2024) [37], the parameters, cell and ethanol yield coefficients (*Y*_*X*/*S*_ and *Y*_*E*/*S*_, respectively), fermentation efficiency (*E_t_*, in %), and volumetric ethanol productivity (*P_E_*, in g·L^−1^·h^−1^) were calculated using Equations (1)–(4), respectively:
(1)$$Y_{X/S}=\frac{C_{Xf}V_{f}-C_{X0}V_{0}}{C_{s0}V_{0}-C_{Sf}V_{f}}$$
(2)$$Y_{E}=\frac{C_{Ef}V_{f}-C_{E0}V_{0}}{C_{s0}V_{0}-C_{Sf}V_{f}}$$
(3)$$P_{E}=\frac{C_{Ef}}{t_{f}}$$
(4)$$E_{t}=\frac{Y_{E/S}}{0.511}\times 100$$
where the subscripts “0” and “*f*” represent the initial and final times of the culture, respectively, and *t_f_* is the final time of the fermentative process. The theoretical ethanol yield coefficient is 0.511 (in gE·gS^−1^).

### 2.5. Analytical Methods

The total solids content (TS) was determined using an infrared dry scale and the lipid content through hexane extraction using Soxhlet equipment. A 160 mL amount of hexane was used on 5 g of sample FW [38]. The same procedure was used for lipid extraction. Extractives and water-soluble compounds were determined by extraction with ethanol and water, respectively. The protein content was determined by the Kjeldahl method [39]. Contents of glucan (cellulose and starch), hemicellulose, ashes, and lignin were determined as described by Gouveia et al., (2009) [40]. The reducing sugars concentration (RS) was measured by reaction with 3,4-dinitrosalicylic acid [41]. The concentrations of glucose, glycerol, and ethanol were measured using high-performance liquid chromatography (Agilent HP 1100 equipped with a refractive index detector (RID)). The samples were analyzed at a flow rate of 0.6 mL min^−1^ at 60 °C using H_2_SO_4_ 5 mM as the mobile phase in an Aminex HPX-87-H column (Bio-Rad, Hercules, CA, USA).

### 2.6. Statistical Analysis

Statistical analyses were performed using Statistca 7.0 software (Pareto chart) and R (Tukey test). The results were analyzed to 95% confidence. Values of *p* < 0.05 indicate non-significant values.

## 3. Results and Discussion

### 3.1. Food Waste Characterization

The FW had a mixed matrix, including grains (beans, rice), pasta, meat, vegetables, fruits, potato, etc., making it a very heterogeneous biomass. This characteristic was due to the FW coming from a university cafeteria. Mixed waste from cafeterias or food courts such as the one used in this study is being analyzed for energy production, such as for biochar obtention [42] and bioethanol production [21,43].

The FW composition had a total solid content (TS) of 15% and 85% moisture (Table 1). The moisture found in this study was slightly superior to that found by Dhakal et al., (2023) [44] and Van Rooyen et al., (2024) [21], whose food waste had 78% and 76% moisture content, respectively. In their recent review, Kannah et al., (2020) [5] stated that food waste moisture content ranges between 50 and 80%, not far from what was found in this work. As FW has a wide variety of compositions, it is safe to say that the moisture found for the FW is reasonable and corroborated by the literature.

Concerning the carbohydrates in the FW, it had a high content of glucans (66.26 ± 8.10%), meaning that there was a predominance of amylaceous substrates in the waste. The total carbohydrate in FW presents a wide content range in the literature, making it possible to find contents in the range of 30–70% [4]. The glucan content found in this study was similar to that found by Hafid et al., (2017) [24] (60.78 ± 2.02%) and Chen et al., (2019) [45].

The protein content of the present FW was also similar to that of the FW used by Hafid et al., (2017) [24] and Kim et al., (2018) [25], which had 20.53 ± 1.77 and 21.5 ± 1.0%, respectively. Proteins can be an issue when hydrolyzing food waste with acid at high temperatures since they provide amino groups that can interact with sugars in a Maillard reaction, producing inhibitory compounds [46].

The FW’s lipid content was 11.78 ± 0.24%. The presence of lipids is a common characteristic in this type of residue, as oils and fats are components of most foods, and are used in meal preparation as well. Similar results were reported in the literature for different types of FW [4]. The presence of high lipid contents can be an issue for FW bioconversion to bioproducts of interest since they can inhibit enzymatic hydrolysis. Amylose, one of the polysaccharides that constitutes starch, bonds to some lipids, forming semi-helicoidal complexes, which increases the structure crystallinity and, consequently, the recalcitrance of the matter [47].

**Table 1 foods-13-04032-t001:** Food waste chemical composition.

	Content (% db *)						
Component	This Study	Matsakas et al., (2014) [48]; Alamanou et al., (2015) [49]	Huang et al., (2015) [50]	Hafid et al., (2017) [24]	Kim et al., (2018) [25]	Carmona-Cabello et al., (2020) [51]	Taheri et al., (2021) [52]
Total solids	15	ND	36	ND	ND	26.1–47.9	90.9 ± 0.1
Water solubles	40.87 ± 0.27	33.81 ± 0.42	ND	ND	ND	ND	35.0 ± 1.9
Total carbohydrate	ND	ND	ND	60.78 ± 2.02	ND	14–20	ND
Glucan **	66.26 ± 8.10	ND	ND	ND	23.1 ± 0.8	ND	ND
Starch	ND	ND	63.5	ND	ND	ND	9.4 ± 0.2
Cellulose	ND	18.30 ± 0.19	ND	ND	ND	ND	9.0 ± 0.3
Hemicellulose	ND	7.55 ± 0.39	ND	ND	ND	ND	9.4 ± 0.2
Pectin	ND	3.92 ± 0.33	ND	ND	ND	ND	ND
Insoluble lignin	2.77 ± 0.31	2.16 ± 0.25	ND	ND	ND	ND	ND
Lipids	11.78 ± 0.24	11.91 ± 0.68	4.1	13.65 ± 1.74	ND	7.2–11.8	11.7 ± 0.4
Proteins	22.85 ± 1.97	10.51 ± 0.37	13.9	20.53 ± 1.77	21.5 ± 1.0	4.6–11.4	13.5 ± 0.4
Ashes	8.63 ± 0.80	11.03 ± 0.42	3.4	ND	16.6 ± 0.5	1.3–2.6	13.2 ± 0.3
Extractives	10.74 ± 0.45	ND	ND	ND	ND	ND	ND

* db = dry basis; ** glucan = cellulose and starch; ND = not determined.

### 3.2. Acid Hydrolysis

The acidic hydrolysis of biomasses frequently uses high temperatures and high acid concentrations. These conditions can be harmful if the hydrolysis aims to use the hydrolysate as a substrate for microbial growth and fermentation. Acids under high temperatures dehydrate carbohydrates, producing furans and organic acids, which inhibit fermentation [53]. Likewise, as previously mentioned, proteins and carbohydrates bond at high temperatures through Maillard’s reaction, creating compounds with a complex structure that is difficult to break. If the reaction occurs above 150 °C, the reaction is irreversible [50]. The FW utilized in this study had a high protein content, 22.85% (Table 1), which is common for food wastes and makes it prone to Maillard’s reaction. Therefore, the chosen temperature for acidic hydrolysis of the FW was 127 °C.

From the conditions evaluated in this study, the highest concentration of reducing sugars (76.14 g·L^−1^) was achieved by applying a higher acid concentration (1.5%), higher solid load (15%), and longer time (60 min) (Table 2). The statistical analysis showed that the acid concentration, the solid load, and their interaction were significant for the recovery of reducing sugars (RS) from the FW (Figure 1). All the single influences were positive, meaning that increasing their values would increase the RS obtention. Not surprisingly, their interaction was also positive, which means that increasing or decreasing both at the same time would lead to higher RS concentrations. In this case, a mutual increase would enhance the hydrolysis results. Nonetheless, increasing solid load may impair mass and heat transfer during the process, diminishing its efficiency [54]. Also, acids with higher concentrations cost more, are corrosive for equipment, and induce the production of furans and organic acids, which inhibit fermentation and microbial growth [53].

The best assay was then repeated in triplicate, reaching 78.69 ± 8.36 g·L^−1^ of RS and an efficiency of 76.26%, which is superior to the efficiency of 48.40% obtained by Kim et al., (2018) [25], who used 0.37% (v·v^−1^) of H_2_SO_4_ at 149.80 °C for 123.6 min, and to the hydrolysis performed by Hafid et al., (2017) [24], which had an efficiency of 42.40% when using 1.5% (v·v^−1^) of HCl at 90 °C for 180 min. The acid hydrolysis in the defatted FW increased the RS release by 6.4% (83.71 ± 2.64 g·L^−1^), which was not significant at a 95% confidence level.

Usually, acidic hydrolysis is used as a pretreatment of lignocellulosic biomasses to remove hemicellulose. In these cases, the yield of RS is lower than the one found in this study, in the range of 2–5% [55]. FW has a lower lignin content (Table 1); thus, the carbohydrates are more accessible to the acid, which makes the process more efficient.

### 3.3. Enzymatic Hydrolysis

The high lipid content in FW (11.78 ± 0.24%) may inhibit enzymatic activity. However, the SR concentration dropped by 2.5% when using defatted FW as a substrate for enzymatic action, which was not significant at a 95% confidence level. Taheri et al., (2021) [52], on the other hand, saw a significant difference (32%) in enzymatic conversion after removing lipids from FW. Their residue had a similar lipid content to the one in this study (11.7 ± 0.7%); however, it had a lower carbohydrate content (26.0 ± 0.8%). Thus, lipid inhibition is probably associated with the ratio of carbohydrate content to lipid content.

The enzymatic hydrolysis had an efficiency of 72.7%, recovering 52.7 ± 3.3 g·L^−1^ of RS. The efficiency was inferior to the one achieved by acid hydrolysis, possibly due to the low enzyme load used. However, increasing the load leads to higher costs since industrial enzymes are expensive.

Taheri et al., (2021) [52] reached 90% efficiency when carrying out simultaneous hydrolysis of defatted FW with amylases and cellulases. The results of this work were superior to the ones found by Alamanou et al., (2015) [43], who used a mix of cellulases and β-glucosidase, achieving 45% efficiency.

### 3.4. Ethanol Fermentation Using Hydrolysate from Food Waste

Acid hydrolysis was more efficient than enzymatic hydrolysis, having a carbohydrate concentration of 77.40 ± 0.70 g·L^−1^ for the untreated medium. As there was no significant difference in lipid extraction, fermentation was continued without the extraction step. Fermentation was carried out using four different strains of *Saccharomyces cerevisiae* for 24 h.

Table 3 details the sugar concentration in the hydrolysate before and after fermentation by each yeast strain, as well as the final concentration of ethanol obtained in the fermentation and ethanol yield coefficient (YE/S). All strains, except for E491, consumed the glucose present in the media in its totality. E491 is an industrial strain which is adapted to lower temperatures, around 17 °C, with fermentation lasting up to 348 h [36]. The strain possibly suffered inhibition due to the high temperature used. The strain TR-313, despite being used in wine production, usually works well under higher temperatures, around 25 °C [35]. Nonetheless, all strains led to bioethanol yields of above 0.40 g·g^−1^, reaching 0.49 g·g^−1^ for JP1, an efficiency of 96%, with no statistically significant difference at a 95% confidence level from the Fleischmann and the TR-313 strain. For this strain, the production was 31.68 ± 0.90 g·L^−1^ of ethanol.

Previous research with FW obtained a bioethanol production of 29.1 g·L^−1^ [56] and 40 g·L^−1^ [57] when carrying out enzymatic hydrolysis followed by fermentation with industrial *S. cerevisiae*. Furthermore, a bioethanol yield of 0.33 g ethanol·g^−1^ glucose was obtained using *S. plantesis* with nutritional supplementation, an efficiency of 85.5% [58]. Concerning the hydrolysates from acid hydrolysis, Kim et al., (2018) [25] obtained 25 g·L^−1^ of bioethanol after fermentation with *I. oriental*. After a sequence of acid and enzymatic hydrolysis, and fermentation with *S. cerevisae* for 24 h, Hafid et al., (2017) [24] achieved 10.92 g·L^−1^ of bioethanol.

The best yield of bioethanol was 0.49 kg·kg glicose^−1^, obtained using the acid hydrolysate of non-defatted FW and the industrial strain JP1 of *S. cerevisiae*. Estimating for 1 ton of FW with the chemical composition described in this work, it would be possible to produce 53 L of bioethanol (Figure 2). In comparison, 1 ton of sugarcane allows the production of 70 to 85 L of bioethanol [59,60]. Therefore, FW would produce 62 to 76.0% of the ethanol obtained from sugarcane whilst being a residue that does not demand land and harvesting processes or deplete food resources.

When thinking about energy transition, it is important to look at local scenarios. In the case of Brazil, the country has a consolidated ethanol industry, which can be improved upon to make this fuel more competitive. However, increasing production would lead to a higher demand for land. The country’s main biomass, sugarcane, has already reached its maximum expansion potential [61]. Currently, the production of corn ethanol is growing in the country [62]. However, corn also demands land for plantations and it is an important food resource in Brazil. In this sense, unconventional biomasses, such as FW, can be an interesting substrate for increasing the production of this biofuel.

## 4. Conclusions

Food waste is one of the main residues produced by society in all steps of the food chain and it causes several social, environmental, and economic problems. Reducing and reusing this residue are important issues for the bioeconomy. In this work, FW was successfully used to produce bioethanol from sugars released after its acidic hydrolysis. The acid hydrolysis process proved to be 5% more efficient than the enzymatic, in addition to being faster and less expensive. The use of Life Cycle Analysis (LCA) tools can help identify the environmental impacts of each technique and it is the next step for this research. Removing lipids from FW before the hydrolysis did not make any difference in sugar release, so the step can be removed from the process and consequently reduce production costs. It was observed that the hydrolysate showed good fermentability when tested with three industrial strains of *Saccharomyces cerevisiae* and a commercial strain of the same yeast. The JP1 strain, used in the bioethanol industry, had the highest yield of 0.49 g·g^−1^ of ethanol. Ethanol production with FW can correspond to up to 76.0% of that obtained with sugarcane, making this substrate interesting in the scenarios of expanding the use of biofuels and energy transition.

## Figures and Tables

**Figure 1 foods-13-04032-f001:**
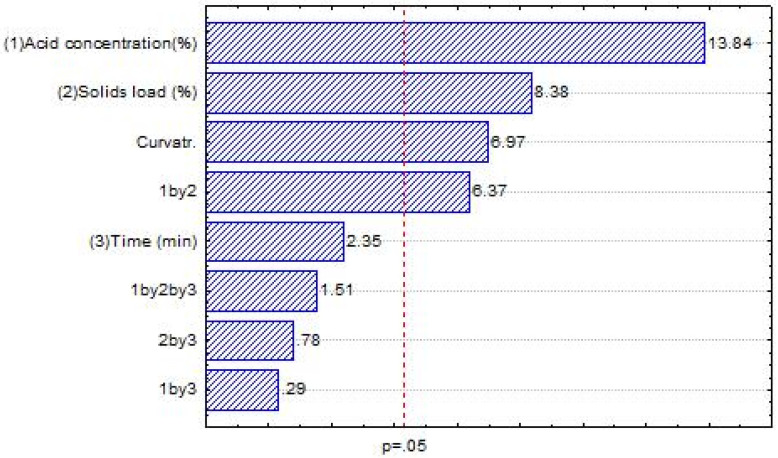
Pareto chart for variable reducing sugar (RS) of experimental planning for acid hydrolysis at constant temperature (127 °C).

**Figure 2 foods-13-04032-f002:**
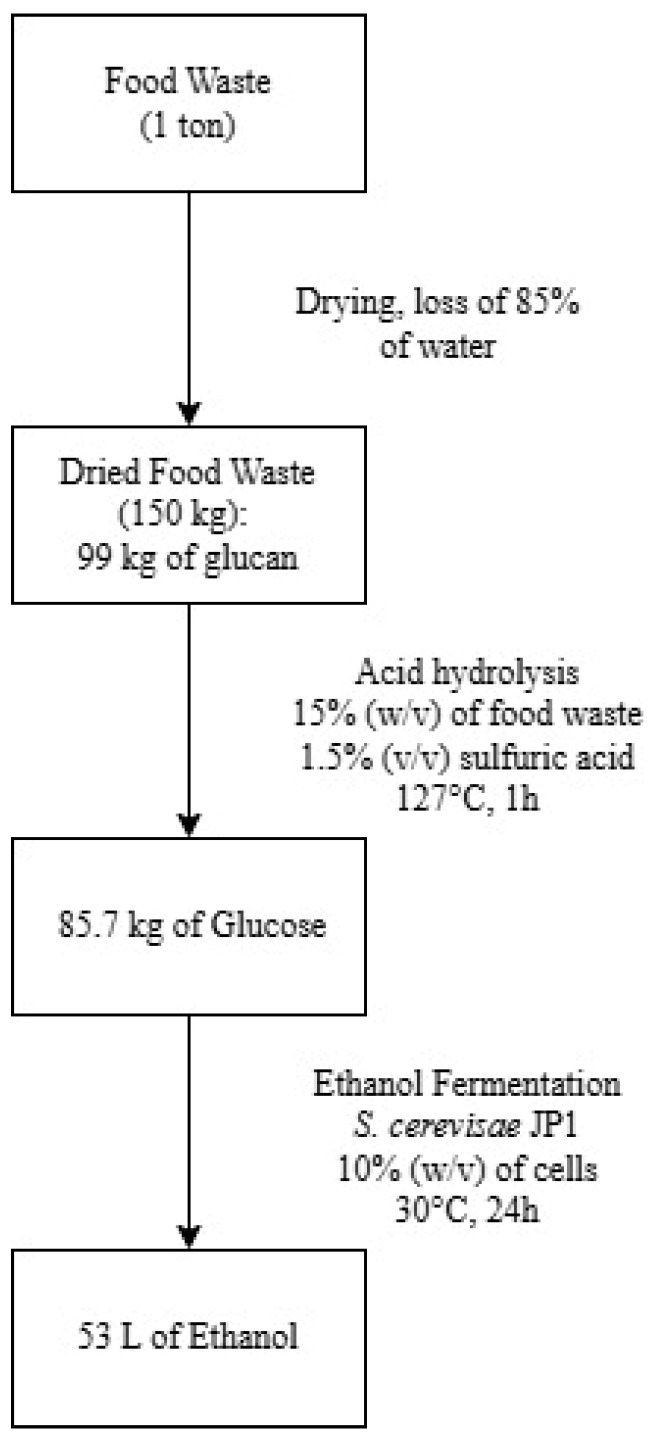
Mass balance for ethanol production from 1 ton of food waste.

**Table 2 foods-13-04032-t002:** Reducing sugars from the acid hydrolysis factorial design at constant temperature (127 °C).

Assay	Acid Concentration (% v·v^−1^)	Solid Load (% m·v^−1^)	Time (min)	RS (g·L^−1^)
1	0.5	5	20	5.94
2	0.5	5	60	13.46
3	0.5	15	20	13.32
4	0.5	15	60	16.88
5	1.5	5	20	29.32
6	1.5	5	60	30.28
7	1.5	15	20	62.85
8	1.5	15	60	76.14
9A	1.0	10	40	52.60
9B	1.0	10	40	49.31
9C	1.0	10	40	45.01

**Table 3 foods-13-04032-t003:** Substrate consumption and bioethanol production by the four yeast strains using the hydrolysates from FW.

	Hydrolysate	TR-313	E491	Fleischmann	JP1
Glucose (g·L^−1^)	65.04 ± 0.16				
Maltose (g·L^−1^)	7.71 ± 0.22				
Glycerol (g·L^−1^)	0.79 ± 0.27				
Consumed carbohydrates (g·L^−1^)	-	65.35 ± 0.12	49.75 ± 2.04	68.22 ± 0.56	65.27 ± 0.25
Final glycerol (g·L^−1^)	-	5.36 ± 0.70	7.59 ± 0.31	5.14 ± 0.73	5.96 ± 0.69
Ethanol concentration (g·L^−1^)	-	30.08 ± 1.35	19.91 ± 0.94	31.61 ± 1.18	31.68 ± 0.90
Ethanol yield (g·g^−1^)	-	0.46 ± 0.02	0.40 ± 0.00	0.46 ± 0.02	0.49 ± 0.02

## Data Availability

The original contributions presented in the study are included in the article, further inquiries can be directed to the corresponding author.

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
