# Peer review of "Brazilian Food Waste as a Substrate for Bioethanol Production"

_foods, 2024, doi:10.3390/foods13244032_

Round 1

Reviewer 1 Report

Comments and Suggestions for Authors

The research topic on the use of waste is current in the context of strategies for the best development of the circular bioeconomy. A good methodological design is followed with adequate experimental methods, originating results that I consider relevant in reference to the potential use of wastewater from agroindustry for industrial purposes, in this case through its recycling for the production of bioethanol.

However, the work has limitations regarding the final approach of the scientific discussion and the conclusions. There is a clear lack of discussion in environmental and sustainability terms (pillars of the circular bioeconomy). From the title, the authors opt for acid hydrolysis with economic justifications, since the biological (enzymatic) method they compared, although it has lower yields, the difference between these methods justifies a more robust discussion in terms of at least proposing a life cycle analysis (LCA). Acid hydrolysis generates waste with a high environmental impact, probably more controversial than the wastewater itself that is the subject of this study.

In this sense, I strongly recommend having a discussion a conclusion with a focus on sustainability and proposing as future work the realization of an LCA and its components. In this sense, the title of the manuscript should be changed to the experimental work and experimental design, and not to the result obtained with acid hydrolysis with its economic justification.

Author Response

1. Summary

Thank you very much for taking the time to review this manuscript. Please find the detailed responses below and the corresponding revisions/corrections highlighted/in track changes in the re-submitted files (text in blue).

2. Questions for General Evaluation

Reviewer’s Evaluation

Response and Revisions

Does the introduction provide sufficient background and include all relevant references?

Can be improved

The introduction has been modified. All changes are in blue in the new file.

Are all the cited references relevant to the research?

Yes

Is the research design appropriate?

Yes

Are the methods adequately described?

Yes

Are the results clearly presented?

Yes

Are the conclusions supported by the results?

Must be improved

The conclusion has been modified. All changes are in blue in the new file.

3. Point-by-point response to Comments and Suggestions for Authors

Comments 1: There is a clear lack of discussion in environmental and sustainability terms (pillars of the circular bioeconomy).

Response 1: Thank you for pointing this out. We added the following passage in lines 55-61, on page 2: “The biotechnological valorization of food waste for the production of biofuels and high-value-added products can contribute to the advancement of the circular bioeconomy, as it represents the reuse of biological resources, increasing the efficiency of the system [13]. Several advantages can be highlighted in the use of FW from the perspective of circu-lar bioeconomy, reducing waste, saving natural resources, improving energy efficiency [14], recovering nutrients and reducing greenhouse gas emissions [15].”

Comments 2: From the title, the authors opt for acid hydrolysis with economic justifications, since the biological (enzymatic) method they compared, although it has lower yields, the difference between these methods justifies a more robust discussion in terms of at least proposing a life cycle analysis (LCA).

Response 2: We thank you for the comment. On page 2, from lines 84 to 89, we added the following passage: “The use of the acid hydrolysis route of food waste has advantages when compared to enzymatic hydrolysis, it tends to be cheaper, since the acids used (sulfuric and hydrochloric) are produced on a large scale and widely available in different regions. In addition, it has faster carbohydrate conversion rates. Enzymes have high specificity, therefore, for a heterogeneous substrate such as FW, acid hydrolysis may be a preferable route.”

Additionally, the title was changed to: “Brazilian food waste as a substrate for bioethanol production”

In relation to LCA, we understand the importance of the methodology, however it was not the objective, at this time, to carry out this study. In the conclusions, however, we add in lines 333 - 335 the following passage: “The use of Life Cycle Analysis (LCA) tools can help identify the environmental impacts of each technique and it is a next step for this research.”

Comments 3: Acid hydrolysis generates waste with a high environmental impact, probably more controversial than the wastewater itself that is the subject of this study.

Response 3: We appreciate the comment. The use of acids for pre-treatment of lignocellulosic biomass generates a wastewater, which has environmental implications. However, in the case addressed in this study, this effluent is being destined for ethanolic fermentation. After fermentation and consequent distillation, a vinasse will be generated, however it is beyond the scope of this study to address this topic. We care about the topic and will certainly study it in the future.

Comments 4: In this sense, I strongly recommend having a discussion a conclusion with a focus on sustainability and proposing as future work the realization of an LCA and its components.

Response 4: We appreciate the comment. The conclusion was modified. The following passages was added in lines 329-330: “Reduce and reuse this residue it’s an important issue regarding of bioeconomy”, and in lines 333-335: “The use of Life Cycle Analysis (LCA) tools can help identify the environmental impacts of each technique and it is a next step for this research.”

Comments 5: In this sense, the title of the manuscript should be changed to the experimental work and experimental design, and not to the result obtained with acid hydrolysis with its economic justification.

Response 5: We agree with the comment. The title was changed for: “Brazilian food waste as a substrate for bioethanol production”.

Reviewer 2 Report

Comments and Suggestions for Authors

In the manuscript of food-3294028, the concept is good. Convering food waste or other MSW into bioethanol is good topic. But the current manuscript do not meet the requirement of good scientific paper. My comments are listed below.

(1) I think the introduction section should make a significant revision, as all statements including backgrounds, current progresses are known to almost all reader. The introduction lacks the statement of innovation, like the current weakness of hydrolysis, differences between each yeast, the entire experiment was design basing on what concern? I think the author should provide what is the scientific/industrial issues in such area, how will this study solve these issues, by solving such issues, what can we benefit from it. That would make the introduction significant.

(2)  Fig 2, from both experimental or industrial view, no significant different.

(3) It is still unclear why the author take 4 type of yeast into comparison, and the reason of different ethanol production is not well discussed.

Hope the author conld consider to revise their work following the above suggestions 

Author Response

1. Summary

Thank you very much for taking the time to review this manuscript. Please find the detailed responses below and the corresponding revisions/corrections highlighted/in track changes in the re-submitted files (text in blue).

2. Questions for General Evaluation

Reviewer’s Evaluation

Response and Revisions

Does the introduction provide sufficient background and include all relevant references?

Must be improved

The introduction has been modified. All changes are in blue in the new file.

Are all the cited references relevant to the research?

Yes

Is the research design appropriate?

Can be improved

Are the methods adequately described?

Yes

Are the results clearly presented?

Can be improved

The results have been modified. All changes are in blue in the new file.

Are the conclusions supported by the results?

Must be improved

The conclusion has been modified. All changes are in blue in the new file.

3. Point-by-point response to Comments and Suggestions for Authors

Comments 1: I think the introduction section should make a significant revision, as all statements including backgrounds, current progresses are known to almost all reader. The introduction lacks the statement of innovation, like the current weakness of hydrolysis, differences between each yeast, the entire experiment was design basing on what concern? I think the author should provide what is the scientific/industrial issues in such area, how will this study solve these issues, by solving such issues, what can we benefit from it. That would make the introduction significant.

Response 1: We appreciate the comment. We made some changes to the introduction; we hope they improved it. Modifications are in blue in the manuscript. We added lines 55-61, on page 2: “The biotechnological valorization of food waste for the production of biofuels and high-value-added products can contribute to the advancement of the circular bioeconomy, as it represents the reuse of biological resources, increasing the efficiency of the system [13]. Several advantages can be highlighted in the use of FW from the perspective of circu-lar bioeconomy, reducing waste, saving natural resources, improving energy efficiency [14], recovering nutrients and reducing greenhouse gas emissions [15]”, also we added the lines 84-89 on page 2, as follows: "The use of the acid hydrolysis route of food waste has advantages when compared to enzymatic hydrolysis, it tends to be cheaper, since the acids used (sulfuric and hydrochloric) are produced on a large scale and widely available in different regions. In addition, it has faster carbohydrate conversion rates. Enzymes have high specificity, therefore, for a heterogeneous substrate such as FW, acid hydrolysis may be a preferable route". Additionally, lines 95-101 have been added, as follows: "The ability to metabolize sugars into complex substrates varies strongly between different yeast strains. Different types of yeast can have different levels of tolerance to factors such as pH, temperature and salt concentration [28;29]. Comparing the fermentative efficiency of different strains makes it possible to identify those that are more efficient in the complete fermentation of food waste hydrolysates, which can increase productivity and reduce costs [30;31]. Recent studies have been carried out to compare and identify the best yeasts for different substrates [32;33]."

Comments 2: Fig 2, from both experimental or industrial view, no significant different.

Response 2: We appreciate the comment. We understand that the figure would not be relevant, as it is clear that there is no significant difference between the samples compared, something highlighted in the text. Therefore, we removed figure 2.

Comments 3: It is still unclear why the author take 4 type of yeast into comparison, and the reason of different ethanol production is not well discussed.

Response 3: We appreciate the comment. Regarding the use of different types of yeast, we have added lines 95-101 on page 2: "The ability to metabolize sugars into complex substrates varies strongly between different yeast strains. Different types of yeast can have different levels of tolerance to factors such as pH, temperature and salt concentration [28;29]. Comparing the fermentative efficiency of different strains makes it possible to identify those that are more efficient in the complete fermentation of food waste hydrolysates, which can increase productivity and re-duce costs [30;31]. Recent studies have been carried out to compare and identify the best yeasts for different substrates [32;33]." and lines 139-141 on page 3: “The yeasts chosen in this study have widespread commercial and industrial use, so it is of interest to identify their fermentability in a heterogeneous medium such as food waste hydrolysate.”

Regarding the difference in ethanol production, only strain E491 had a significant difference with the others, as explained on page 9, lines 287 and 288. However, we added the passage " The strain possibly suffered inhibition due to the high temperature used" in lines 288 and 289. We also added the information that no there is a significant difference between the other strains, on page 9, lines 292 and 293: "with no statistically significant difference at 95% confidence level with Fleischmann and the TR-313 strain."

Round 2

Reviewer 1 Report

Comments and Suggestions for Authors

The authors followed the recommendations of the first review, in which the focus of the work was improved, so it could be accepted for publication.

Reviewer 2 Report

Comments and Suggestions for Authors

I think the authors has well addressed the provious problems. It is OK to be accepted.